# Information Processing and Overload in Group Conversation: A Graph-Based Prediction Model

## Gabriel Murray

Department of Computer Information Systems, University of the Fraser Valley,
Abbotsford, BC V2S 7M8, Canada; gabriel.murray@ufv.ca

**Abstract:** Based on analyzing verbal and nonverbal features of small group conversations in a task-based scenario, this work focuses on automatic detection of group member perceptions about how well they are making use of available information, and whether they are experiencing information overload. Both the verbal and nonverbal features are derived from graph-based social network representations of the group interaction. For the task of predicting the information use ratings, a predictive model using random forests with verbal and nonverbal features significantly outperforms baselines in which the mean or median values of the training data are predicted, as well as significantly outperforming a linear regression baseline. For the task of predicting information overload ratings, the multimodal random forests model again outperforms all other models, including significant improvement over linear regression and gradient boosting models. However, on that task the best model is not significantly better than the mean and median baselines. For both tasks, we analyze performance using the full multimodal feature set versus using only linguistic features or only turn-taking features. While utilizing the full feature set yields the best performance in terms of mean squared error (MSE), there are no statistically significant differences, and using only linguistic features gives comparable performance. We provide a detailed analysis of the individual features that are most useful for each task. Beyond the immediate prediction tasks, our more general goal is to represent conversational interaction in such a way that yields a small number of features capturing the group interaction in an easily interpretable manner. The proposed approach is relevant to many other group prediction tasks as well, and is distinct from both classical natural language processing (NLP) as well as more current deep learning/artificial neural network approaches.

**Keywords:** group interaction; information overload; network models; natural language processing; social signal processing; multimodal interaction

---

## 1. Introduction

Being able to tell whether a small group is making full use of available information, or whether they are experiencing information overload, is a valuable skill for a team leader or manager. Information overload—receiving too much information—is associated with lower job satisfaction and higher stress levels [1]. Being able to automatically detect these information processing states would be a useful feature of a virtual meeting assistant. Such a system could be utilized in real-time during a meeting in order to improve information processing and group effectiveness, or offline as a way of tracking team dynamics or auditing a decision that was made [2].

The approach taken in this work to detecting information processing states uses graph-based social network models for representing both verbal and nonverbal aspects of small group interaction. Features are derived from both graph representations and used to predict two outcome variables: whether the group members (in aggregate) felt that they made good use of all available information during their meeting, and whether they suffered from information overload during the meeting.

On the information use prediction task, the combined verbal and nonverbal features yield predictive performance that is significantly better than the performance of baselines that predict the mean or median values from the training data. On the information overload task, the best predictive model does not yield statistically significant improvement over the baselines. We also find that there are no statistically significant differences in using just language features, just turn-taking features, or using the full feature set, and that using only language features gives comparable performance to using the full feature set.

Our goal in this work, beyond the immediate prediction tasks, is to find a representation of conversations that utilizes few features and furthermore yields features that are interpretable. Using few features is important because our predictions are at the level of an entire conversation, and having each conversation constitute a single observation mean that we have very few observations and could easily end up with far more features than observations (known as the $p >> n$ problem, where $p$ is the number of features and $n$ is the number of observations). The reasons for using interpretable features are two-fold. First, from a research perspective, we want our learned models to shed light on how language is used in group interaction. Second, and further to the first point, if we make group-level predictions such as the information processing predictions in this work, or other prediction problems such as group sentiment or group task performance, those predictions should be *auditable* and potentially *actionable* by the group or its leader. For example, if we predict that a group is experiencing information overload, the team leader may want to know which aspects of the group interaction are contributing to the overload, and whether any of them can be changed to alleviate the overload problem. The second problem is difficult, as it requires learning causal relationships between the features and outcomes. We do not address causal modelling in this work. We take initial steps towards representing language in interaction via graph-based social network models, and derive a small number of interpretable features summarizing the conversations.

The approach we take towards achieving the goal of having a small number of interpretable features is distinct from both "classical NLP" approaches as well as more current artificial neural network/deep learning approaches to NLP. The challenges of interpreting deep neural models are well-known [3,4], and there is the additional challenge of trying to use deep learning methods with a very small number of observations, when the availability of plentiful training data is often a key to the success of deep learning [5]. However, the classical approach to NLP also poses a challenge for attaining our goals. For example, learning that there is a relationship between a group outcome variable and the presence/absence of a particular part-of-speech tag or dependency relation may not provide any useful insight about the group interaction dynamics and conversation. The classical NLP approach also tends to use very large numbers of sparse features, again posing a major problem given our very small number of observations.

In contrast to either of the above approaches, we aim to capture group conversation using graph-based representations and to derive a small number of features that may relate to group outcomes of interest. We see this approach to representation and feature learning as being relevant to many other group prediction tasks beyond the information processing tasks covered in this work.

The structure of this paper is as follows. In Section 1, we discuss related work on natural language processing and social signal processing for small groups, as well as work on virtual meeting assistants, and research on information load in other domains. In Section 2, we present the graph-based social network models used to capture verbal and nonverbal aspects of group interaction. Section 3 describes the experimental setup, including the meeting dataset, outcome variables, and evaluation metrics. The main results are presented in Section 4, and we conclude in Section 5, along with thoughts on future work.

*Related Work*

Recent work by Lai and Murray [6] aims to automatically predict group satisfaction in meetings, using both verbal and non-verbal features. In that work, three aspects of satisfaction are predicted:

overall satisfaction, attention satisfaction (whether members felt they received sufficient attention), and information overload. In our current work, we also predict information overload amongst group members, as well as an additional measure of group satisfaction: whether the members felt that they made good use of all available information. Like Lai and Murray, we utilize both verbal and non-verbal features. However, their approach is more similar to classical NLP, using features such as part-of-speech tags, dependency relations, and bag-of-words features, while we use graph-based social network representations of each conversation to derive a small set of interpretable features for machine learning. Other recent work on predicting group affect involves graph-based representations of turn-taking for modelling the relationship between group interaction and group satisfaction measures, but does not include verbal features [7].

Another related vein of research is on detecting sentiment or subjectivity in individual sentences or dialogue act units within meetings [8]. In contrast, we are not making sentence-level predictions but rather predictions for the entire meeting; specifically, whether the participants felt that they were making good use of all available information in the meeting, and whether they experienced information overload during the meeting.

Other work applying machine learning to group interaction includes work on automatic prediction of group performance on a task, using nonverbal features [9] or a combination of verbal and nonverbal features [10], and detection of emergent leadership in groups [11,12].

Automatic meeting summarization [13,14] also relates to the current work, as one goal of meeting summarization is to reduce information overload during or after meetings. More generally, our work relates to projects on developing intelligent virtual meeting assistants [15,16], as we see dealing with information overload as an important capability of such agents. Even more broadly, there has been some research outside the meetings domain on using machine learning and artificial intelligence to help alleviate information overload in different application areas [17–19].

## 2. Social Network Models

In this section we describe two graph-based social network representations for modelling verbal and nonverbal aspects of group interaction, respectively. In both cases, the goal is to represent aspects of group interaction so that we can extract a small number of interpretable features of the conversation. Features are extracted from both the verbal and nonverbal social network models and combined to yield a multimodal perspective on the group interaction.

These experiments use the AMI meeting corpus [20], where each meeting features four people role-playing as a team designing a product (a remote control). Each group completes a series of four meetings. Each of the participants is assigned a distinct role, as a project manager, marketing expert, industrial designer, or user interface designer.

The first step in building all of the graph models is to order a meeting's dialogue act units by start time, and then segment each meeting into non-overlapping windows of 20 dialogue act units each. Choosing 20 dialogue act units as the window-size is motivated by the fact that it is sufficiently large that it will likely feature multiple participants within each window, and small enough that each window will foreground a small portion of a much larger conversation. The average number of dialogue acts per meeting in this subset of the AMI corpus is approximately 980, though the initial group meetings can be much shorter (200–300). While we use the dialogue act *segmentations* provided in the AMI corpus, the dialogue act *types* (e.g., question, inform, backchannel) are not used at all in these experiments.

### 2.1. Language Graphs

Within each window of 20 dialogue act units, a language graph is built that represents the group conversation up to and including the current window. Conceptually, this is a tripartite hypergraph, where each edge connects three nodes representing three types of entities: words (i.e., word *types*, as opposed to word *tokens*), dialogue act units, and speakers. A given edge therefore represents a word

used in a particular dialogue act unit by a particular speaker. The left side of Figure 1 shows two toy sentences, each associated with a different speaker. There are nodes corresponding to the two speakers, the two sentences, and the distinct word types used in the sentences. Each of the edges (the dashed lines) connects three nodes: a speaker, a sentence, and a word type.

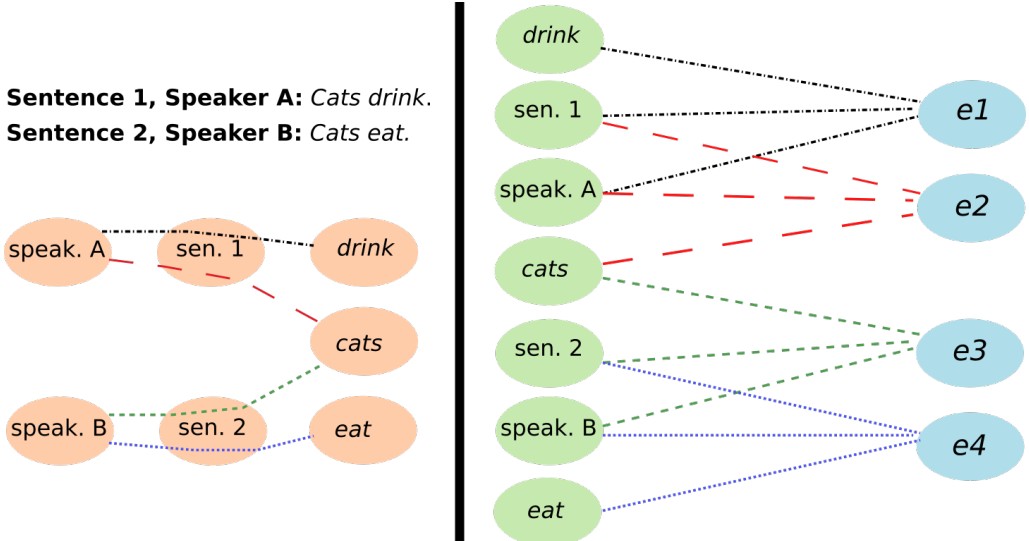

**Figure 1.** Sample Graph Conversion. *Left Side:* Sample Sentences and Corresponding Tripartite Hypergraph *H*; *Right Side:* Corresponding Bipartite Graph *B*.

This tripartite hypergraph as just described can also be represented as an equivalent bipartite graph. In these experiments, we utilize the bipartite representation for extracting features from the graphs. The tripartite graph *H* is transformed into a bipartite graph *B* as follows. The bipartite graph *B* has partitions *N* and *E*, where the nodes of *N* are the nodes from the original hypergraph *H*, and the nodes of *E* are the edges of the original graph *H*. The resulting graph *B* will contain an edge $< n_1, e_1 >$ connecting nodes of *N* and *E* if $n_1$ belonged to edge $e_1$ in *H*. The right side of Figure 1 shows the equivalent bipartite graph for the tripartite graph shown earlier. The left portion of the bipartite graph contains all of the nodes from the original graph, while the right portion contains nodes that were edges in the original graph.

Figure 2 shows an example bipartite language graph generated partway through one of the meeting discussions used in these experiments. The nodes representing the four participants are highlighted, as are the edges connected to those four nodes. The nodes in the left partition that are not highlighted correspond to the words and dialogue act nodes from the original graph *H*, while the nodes in the right partition correspond to edges in *H*.

Many of the features we extract from the language graph involve the betweenness centrality of each participant in the language graph (the *linguistic centrality*, for short). However, this centrality measure will tend simply to correlate with the frequency of the participant's dialogue acts in the conversation up to that point. For example, the participant represented by the dark node in the lower-left of Figure 2 has participated less often in the conversation so far, and will likely have a lower betweenness centrality as a result. For that reason, we divide the centrality measure of each participant by their participation frequency, i.e., by the proportion of dialogue act units in the conversation so far that belong to them.

Within each 20 dialogue act window, we calculate the linguistic centrality of each speaker, rescaled by that speaker's participant frequency, as just described. These are then averaged over all of the conversation windows, for each participant. Finally, we extract the following features relating to language graph centrality:

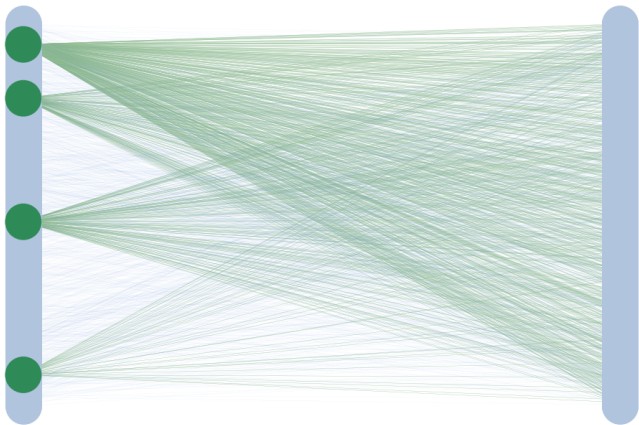

**Figure 2.** Bipartite Language Graph with Participant Nodes/Edges Highlighted. *Left Side:* Nodes in *B* that were also nodes in the original hypergraph *H*; *Right Side:* Nodes in *B* that were edges in the original hypergraph *H*.

- The minimum, maximum, and mean of the averaged linguistic centrality scores (*min_lingcent*, *max_lingcent*, *mean_lingcent*).

Similar centrality features could be extracted by building a single graph for the entire conversation and calculating the centrality of each participant, rather than using the window-based approach as just described. The two approaches could give differing centrality results in some cases, however. For example, a participant might become very central late in the meeting, after not being central for most of the meeting up until then—perhaps they were listening for most of the meeting, and at the end of the meeting are agreeing with or reiterating things that others have already said. A single conversation graph might show them as very central, while the window-based approach will yield a lower averaged score.

Other features involve community detection (or clustering) on the language graph. We use modularity-based community detection [21], where each node begins in a singleton community, and each step consists of merging two communities that most increase the modularity, until the modularity cannot be improved. One advantage of this approach is that the number of communities does not need to be pre-specified, and can be used as a predictive feature. Because the original graph is a tripartite hypergraph, each of the resulting communities can contain a mix of participant nodes, sentence nodes, and word nodes. A qualitative analysis of these communities shows that there is typically a small number *n* (where *n* ≈ the number of participants) of large communities where each of these large communities roughly corresponds to the vocabulary of a particular participant. There is then a larger number of small communities, each of which roughly corresponds to a topic in the discussion. Recent research [22] shows the advantages of taking a graph-based approach to topic detection and also describes potential weaknesses of popular alternative approaches such as Latent Dirichlet Allocation (LDA) [23]. However, our current work differs from that recent research in that our graphs include not only words and sentences but participants as well, and so the resulting community structure relates not only to topics but also to lexical similarities and differences between the conversation participants.

The features that relate to community structure are as follows. For each 20 dialogue act window, we calculate the total number of communities in the language graph for the entire conversation up to that point. We also calculate the number of communities represented by sentences in the current window. This will range from 1 (all sentences in the current window belong to one community) to 20 (each sentence in the window is in a different community). We further derive the current number of communities containing speakers. For this corpus, this feature will range between 1 (all speakers in the

same community) to 4 (each speaker in a separate community). All three of those community-based features are calculated at each window in the conversation. We then calculate the following summary statistics over all of the windows in the conversation to create the following features:

- The minimum, maximum, and mean number of communities (*min_num_comms*, *max_num_comms*, *mean_num_comms*).
- The minimum, maximum, and mean number of communities represented by sentences in each window (*min_num_win_comms*, *max_num_win_comms*, *mean_num_win_comms*).
- The minimum, maximum, and mean number of communities containing speakers (*min_num_speak_comms*, *max_num_speak_comms*, *mean_num_speak_comms*).

We also extract the following feature relating to the changing language network. For each window, we calculate the change in density of the language network between the previous and current windows. Density is a measure of how connected the graph is, e.g., a graph with the maximum possible number of edges has a density of 1. We then calculate the following summary statistics over the entire conversation:

- The minimum, maximum, and mean language network density change (*min_density_change*, *max_density_change*, *mean_density_change*).

*2.2. Turn-Taking Graphs*

Within each window of 20 dialogue act units, we represent the group interaction dynamics in that window as a directed graph, with nodes representing participants. There is an edge $(A, B)$ between participants $A$ and $B$ if there is at least one immediate transition within that window from $A$'s speaking turn to $B$'s speaking turn. The edge $(A, B)$ has a cost which is the reciprocal of the number of times that transition was made within the current window; i.e., the more we see the transition from $A$ to $B$, the lower the distance between them.

After building the graph for the current window, we extract betweenness centrality, closeness centrality, and degree centrality features for each participant. These centrality measures are then averaged over all the windows in the conversation for each participant. Finally, we take the maximum, minimum, and mean of the averaged centrality scores over all of the participants.

This results in the following 12 turn-taking features (with abbreviations, for later reference):

- The minimum, maximum, and mean of the averaged closeness centrality scores (*min_close*, *max_close*, *mean_close*).
- The minimum, maximum, and mean of the averaged degree centrality scores (*min_deg*, *max_deg*, *mean_deg*).
- The minimum, maximum, and mean of the averaged betweenness centrality scores (*min_bet*, *max_bet*, *mean_bet*).

All graphs were built using networkX (https://networkx.github.io/), and the features were extracted using networkX functions for node centrality, community detection, and network density.

*2.3. Frequency Features*

Finally, we extract the remaining features that are not derived from either of the above graph-based models. Within each 20 dialogue act window, we extract the frequency (number of dialogue acts) for each participant in that window. These are then averaged over all of the windows of the conversation for each participant. We then calculate three features:

- The minimum, maximum, and mean of the averaged frequency scores (*min_freq*, *max_freq*, *mean_freq*).

## 3. Methods and Materials

After each meeting in the AMI corpus, the participants filled out a questionnaire, which included ratings of various satisfaction criteria. In this work, we focus on the follow two criteria, which are rated by each participant on a seven-point Likert scale:

- **Q5**. *Information Use*

  - "All available information is being used."

- **Q15**. *Information Overload*

  - "There was too much information."

Like Lai and Murray [6], we sum the individual ratings over the entire group, meaning there is a maximum possible score of 28 for each criterion. While it might be expected that the two criteria would be inversely related, there is in fact no significant correlation between the ratings (Spearman's rho = $-0.12$, p = 0.20), as shown in Figure 3. It is possible, for example, to feel a sense of information overload but still feel that you are making good use of available information, or to not have information overload but feel (for other reasons) that you did not make good use of available information.

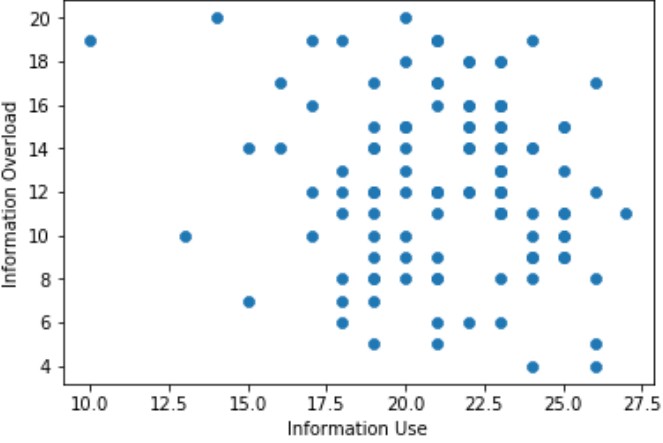

**Figure 3.** Lack of Correlation Between Outcome Variables

The number of observations is 120, which is the number of AMI meetings for which we have these ratings. There are 27 total features. We experiment with linear regression (LM), random forests (RF), and gradient boosting (GB) machine learning models. Due to the small number of observations, we utilize leave-one-out cross-validation. For each outcome variable, we compare the machine learning models with baselines that predicts the mean or median value of the outcome variable in the training folds. For evaluation, we use mean-squared error (MSE).

## 4. Results

The main set of results for both prediction tasks is shown in Table 1. One initial finding is that the *information use* prediction task is easier than the *information overload* task, as exhibited by the overall lower MSE scores. The best model for the information use prediction task is random forests, and according to the two-sided Wilcoxon test, it is significantly better than the mean baseline ($p = 0.0043, W = 2541, N = 120$), the median baseline ($p = 0.0355, W = 2827, N = 120$), and the linear regression model ($p = 0.0019, W = 2442, N = 120$). On the information overload prediction task, random forests is again the best model, and is significantly better than both gradient boosting ($p = 0.0001, W = 2148, N = 120$) and linear regression ($p = 0.0013, W = 2400, N = 120$). There are no significant differences when comparing random forests with the baseline models on that task.

On the information overload task, our best score of 13.38 is worse than Lai and Murray's best reported score of 11.65 [6]. However, they use a much larger feature set consisting of hundreds of acoustic, lexical, and turn-taking features (with feature selection), while the current experiments use only 27 features that relate to language in interaction. While Lai and Murray did not report results on the information use task, we extended their experiments by using their same set of acoustic and lexical features to predict this criterion, and the best performance was an MSE of 6.73 using random forests, outperforming our best model on that task (6.99) by a small margin. This provides encouragement that a small number of well-motivated features can work well on this task.

**Table 1.** MSE Scores for Models and Baselines.

| Model | Inf. Use | Inf. Overload |
|---|---|---|
| baseline (mean) | 8.96 | 15.12 |
| baseline (median) | 9.30 | 15.03 |
| LM | 9.07 | 18.52 |
| GB | 7.64 | 17.86 |
| RF | **6.99** | **13.38** |

Figure 4 shows the top 10 important features for each task, where feature importance is determined by the average reduction in MSE when a feature is used as a split point in a decision tree in the random forests model. For each task, the top features include both verbal and nonverbal predictors. For predicting information use scores, the most important individual feature is *mean_density_change*, i.e., the average change in density of the language network when comparing the density of each window with the density of the previous window. As Figure 5 shows, there is a moderate and significant negative correlation ($r = -0.355$, $p = 0.000068$) between mean density change and the overall information use scores. Several factors can cause large increases in language network density from one window to the next. However, meeting length plays a major role. Network density almost always increases from one window to the next, but the increases get smaller as the meeting goes on. This is due in part to the fact that, in the AMI corpus at least, the team leaders are very active and dominant at the beginnings of meetings, and less so later on. So longer meetings tend to have lower values for *mean_density_change* and higher ratings for information use. Increases in network density may also relate to linguistic entrainment or alignment [24,25], wherein group members begin to use similar language as each other through the conversation.

The second most useful feature is *max_close*, or the maximum of the closeness centrality score of the participants, and it also has a negative correlation with the information use ratings. This indicates that in meetings with more central/dominant participants, group members rate their use of available information as being poorer. Indeed, this is supported by the fact that ratings of information use and attention satisfaction have high positive correlation. In other words, in cases where all participants felt that they contributed to the discussion, they also rated their use of information more highly.

For predicting information overload, the single most useful feature is *mean_num_win_comms*, the average number of communities per window. As Figure 6 shows, there is a negative correlation with this feature and the overall information overload score ($r = -0.446$, $p = 0.00000032$). That is, as the average number of communities per window increases, people are less likely to report information overload. Since communities represent not only topics but also participants, this finding indicates that when many people tend to be active in any given slice of the conversation, then the participants are less likely to feel information overload.

Table 2 shows the coefficient values from the multivariate linear regression model for all of the top features that are shown in Figure 4.

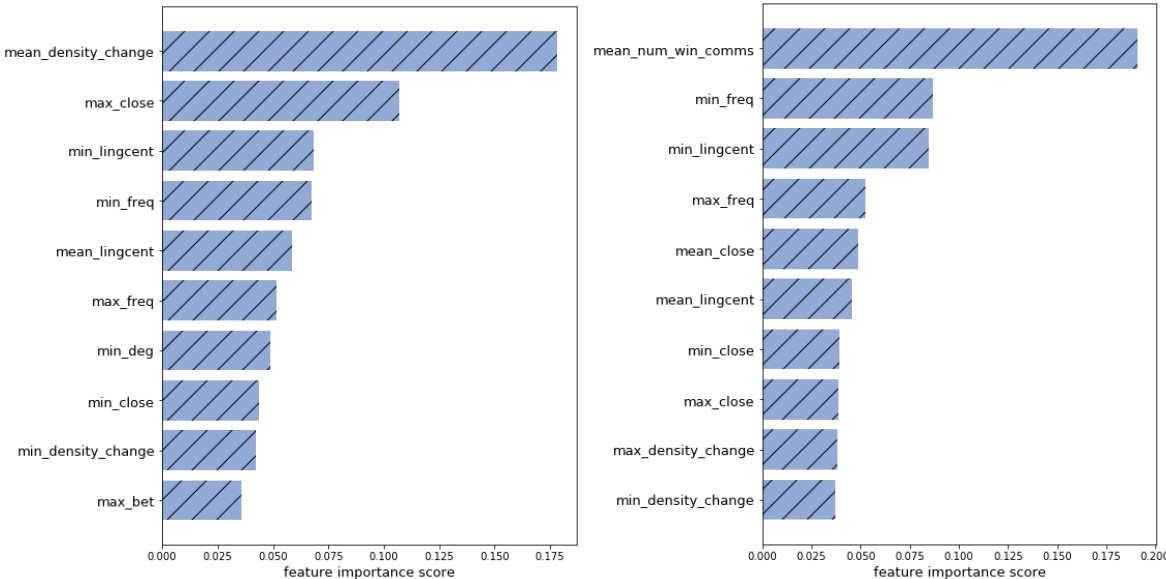

(**a**) Feature Importance—Information Use

(**b**) Feature Importance—Information Overload

**Figure 4.** Feature Importance Scores for Both Tasks.

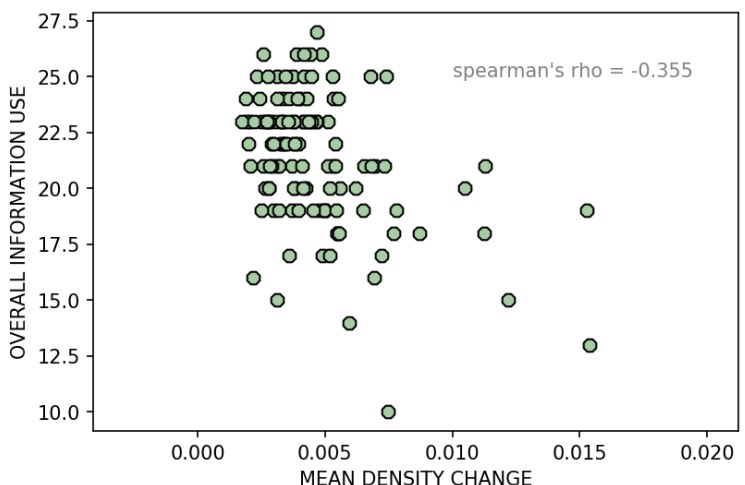

**Figure 5.** Network Density Change vs. Information Use Score.

**Table 2.** Multivariate LR Coefficients for Top Features.

| Feature | Inf. Use | Inf. Overload |
|---|---|---|
| mean_density_change | $-529.578$ | $-166.207$ |
| min_density_change | $14.167$ | $158.908$ |
| max_density_change | $-7.297$ | $-3.480$ |
| mean_lingcent | $3.707$ | $-3.970$ |
| min_lingcent | $-2.196$ | $6.636$ |
| mean_num_win_comms | $0.677$ | $-1.753$ |
| mean_close | $51.277$ | $31.972$ |
| min_close | $-27.359$ | $-99.567$ |
| max_close | $-11.310$ | $-41.714$ |
| min_freq | $-0.014$ | $0.510$ |
| max_freq | $-0.432$ | $0.711$ |
| max_bet | $3.110$ | $-16.276$ |
| min_deg | $8.205$ | $12.194$ |

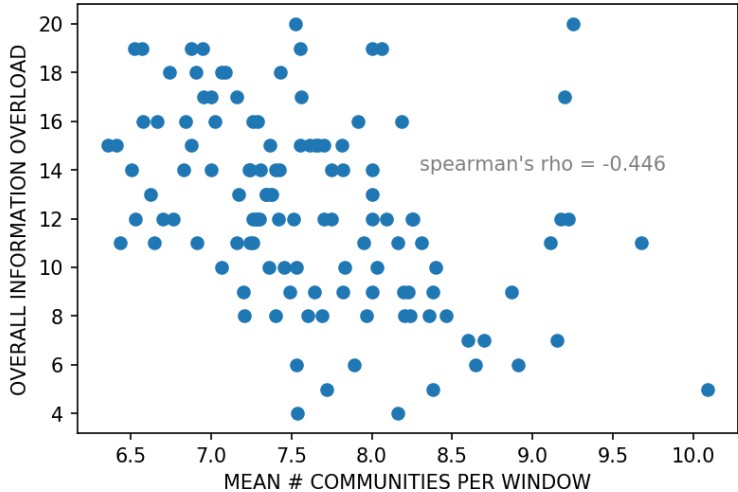

**Figure 6.** Number of Communities vs. Information Overload Score.

*Feature Classes*

Finally, we try both prediction tasks using only linguistic features and using only turn-taking features. Table 3 shows the best MSE scores for each feature class across all machine learning models, compared with the best MSE when using all features. In both cases, the best prediction performance is found by using the full set of verbal and nonverbal features. However, the linguistic feature class outperforms the turn-taking feature set when used independently, and yields competitive results with the full feature set. There are no statistically significant differences.

**Table 3.** MSE: Random Forests Models with Different Feature Classes.

| Feature Class | Inf. Use | Inf. Overload |
|---|---|---|
| Linguistic Only | 7.48 | 13.99 |
| Turn-Taking Only | 8.03 | 15.69 |
| All Features | **6.99** | **13.38** |

## 5. Conclusions

In this work, we have aimed to automatically predict group attitudes towards information processing in a meeting. Specifically, models utilizing verbal and nonverbal features of the conversation were used to predict whether the group members felt that they were making use of all available information, and whether they had experienced information overload during the meeting. The linguistic features were derived from a tripartite hypergraph connecting words, sentences, and speakers. Turn-taking features were extracted from a graph representing turn-taking transitions in a meeting. Performance on the first task was better than on the second, with the random forests model trained on all features yielding the best performance, and significantly outperforming the baselines. Scores overall were worse on the second task, and the random forest model outperformed the baselines, but with no statistically significant difference. We analyzed some of the most useful features for each task, as well as testing the performance of the linguistic feature class and turn-taking feature class on their own. The combined feature set gave the best performance on both tasks, albeit not with significant improvement over either individual feature subclass. Using just linguistic features was particularly effective for both tasks.

A more general goal of this work was to represent conversations in such a way that we could extract a small number of effective and interpretable features for group-level prediction tasks. The proposed approach differs from feature extraction in classical NLP as well as feature learning in neural network approaches to NLP. We see this method of conversation representation and feature derivation as being potentially useful for other group-level tasks as well. The current set of results

contained herein are encouraging, given that the best models used only 27 features of the conversation. In future work, we will investigate additional features that summarize aspects of the conversation graphs, while still aiming to keep the overall number of features low. This is crucial for tasks such as this where the number of observations is small. In future work, we will also look at methods for visualizing the verbal and nonverbal conversation graphs and their derived features, including animated visualizations that show how the structure of the graphs change throughout the meeting.

**Funding:** This research was funded by the Natural Sciences and Engineering Research Council of Canada, grant number 06806.

**Acknowledgments:** Thank you to the anonymous reviewers for very helpful feedback.

**Conflicts of Interest:** The author declares no conflict of interest.

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
