# Peer review of "Information Processing and Overload in Group Conversation: A Graph-Based Prediction Model"

_mti, doi:10.3390/mti3030046_

Round 1

Reviewer 1 Report

The aim of this paper is to design high-level and interpretable features from both verbal and non-verbal (turn-taking) aspects of the conversation to predict the perceived team outcomes of information use and information overload. I think the intuition or reason for some of the design decisions is not clear and should be explained.

Language graphs:

1. It would be good if the author uses a simple concrete example of a hypergraph with only 2 edges and the transformed simple graph. The hyper-edges can be illustrated using a triangle plane.

2. What are the dialogue act types in this corpus? And what is the design decision behind considering dialogue act units as nodes? Why not just words and speakers? The graph could be a simple bipartite graph where two sets of nodes are words and speakers.

3. What is the intuition behind choosing betweenness centrality not other forms of centrality like closeness on language graph?

Turn-taking: graphs:

4. By considering non-overlapping intervals, if we consider a small window size, we miss a lot of interactions in the boundaries. It is not clear if 20 is small for this dataset since the author did not provide any statistics on the average of the total number of dialogue act units in meetings.

If this is the case that 20 is small, it might be a better choice to consider an overlapping window with only one overlapping unit. The author should discuss which is a better choice here.

5. How did the author handle overlapping turns?

6. Why build the graph for each window and then average the centrality features? Why not build the graph for the whole conversation and extract centrality features from it? Should be explained. 

Methods and materials:

 7. Statistics on data that needs to be reported: length of each meeting in terms of dialogue act units, dialogue act types, distribution of the output measures.

8. Mean, median, or mode all could be used as majority baseline. Why did the author choose mean? Is mean the best majority baseline on this data?

9. I would like to see the comparison of this model with the best comparable previous work as a baseline, similar to what the author mentioned in line 218 but on both tasks (the author should extend the experiments in [3] and use the features to predict both outcomes).   Also, since this dataset has only 120 samples, hundreds of features of the model from [3] make the model prone to overfitting. The author should mention if feature selection was implemented for this baseline from [3] to handle this issue. 

10. Are the best results in table 2 significantly better than linguistic only?

11. The discussion of the two most useful features is very interesting. The author should add a table to show the coefficient values for all the features in figure 3. So, we have a better insight on the direction of the relation of these features (besides the two that were discussed in details) with the outcomes. 

12. The author's goal is to design high-level interpretable features that can predict team outcomes. The author in line 233 mentioned how one of the features might possibly be related to entrainment. But, they did not use entrainment as one of the features in this work. I think using entrainment as a feature is in-line with the goal of this paper. This is a suggestion and here are some of the reasons:

    12.1. It can be easily computed at the conversation level.

    12.2. It can be calculated not only on lexical features but also on the acoustic-prosodic level. Considering features from acoustic-prosodic level has shown to be useful in both [3] and [22].

    12.3 It is a high-level interpretable feature (as the goal of this paper) and has been shown to be predictive of several team outcomes. 

13. Some of the Writing issues:

Line 47: "a small number number of"

Line 94: "areas[14-16]"

Line 207: "random forest and gradient boosting" (mention regressors)

Line 266: "best pest performance"

Line 277: "encouraging, give that"

Overall: Add citation for all centrality measures

Reviewer 2 Report

My major concern is, that the authors claim a significance in the abstract and text which is (from my point of view) not shown and even not supported by the presented results.

Related to that the 2 result tables show different values for each case, but I would expect at least for one case identical results. That could be a typo but also an issue of the method.

These issues must be solved/clarified before an acceptation of the article.

In the following I give some hints/questions related to line numbers:

Line 6: The claim is from my point of view not supported by the results. Additionally, the   baseline is addressed several times in the article before it is explained. That should be changed.

Line 13: “neural approaches” means artificial neuronal networks (ANNs)?

20: “Being to” -> “Being able to”?

35:  I was confused from “the p >> n problem” term until I got it. Maybe write in words what you mean.

47: “number number” -> “number”

48: again “neural approaches”

49+50: Please give references to the “well-known” challenges and deep learning methods that try to handle very small numbers of observations.

84: some of the or all available information?

104: please be explicit “words” or “word types” and give examples

Figure 1: Without further explanation, labels and/or examples the figure is meaningless

 Line 114–188: How are the features computed? Details or even formulas would be very helpful.

143: I’m unsure: Does a window contain the last 20 dialogue acts in the dialogue or of a person?

165: Why do you use the reciprocal and not the number itself?

184: “dialogue cat” -> “dialogue act”

185: How many dialogue types do you used, what is contained in one dialogue act? Just the form or also the information transferred?

226/227 + Figure 4: I cannot see a significance? Please explain it or remove the claim.

215: How are the means computed over the different folds? One-sided or two-sided t-test? Are the requirements (normal distribution and equal variance) for a t-test fulfilled? Please read how to report t-test results, e.g. according to APA style.

Table 2.: What are the “best models”? Is it RF? More important: As Table 1 and 2 show the results for “best models” I would expect that some numbers (probably for the “all features row in Table 2) in Table 1 and 2 or identical, but not any is.
